# Assessing $K_{\mathrm{DP}}$-based QPE for the record-breaking rainfall over Zhengzhou city on 20 July 2021

Haoran Li[1,2], Dmitri Moisseev[2], Yali Luo[1,3], Liping Liu[1], Zheng Ruan[1], Liman Cui[4], and Xinghua Bao[1]

[1]State Key Laboratory of Severe Weather, Chinese Academy of Meteorological Sciences, Beijing, China
[2]Institute for Atmospheric and Earth System Research / Physics, Faculty of Science, University of Helsinki, Finland
[3]Collaborative Innovation Center on Forecast and Evaluation of Meteorological Disasters, Nanjing University of Information Science and Technology, Nanjing, China
[4]Henan Meteorological Observatory, Zhengzhou, China

**Correspondence:** Haoran Li (lihr@cma.gov.cn)

**Abstract.** Although radar-based quantitative precipitation estimation (QPE) has been widely investigated from various perspectives, very few studies have been devoted into extreme rainfall QPE. In this study, the performance of $K_{\mathrm{DP}}$-based QPE during the record-breaking Zhengzhou rainfall event occurred on 20 July 2021 is assessed. Firstly, the OTT disdrometer observations are used as input to T-matrix simulation and different assumptions are made to construct $R(K_{\mathrm{DP}})$ estimators. Then,

$K_{\mathrm{DP}}$ estimates from three algorithms are compared for obtaining best $K_{\mathrm{DP}}$ estimates, and gauge observations are used to evaluate the $R(K_{\mathrm{DP}})$ estimates. Our results in general agree with previous known-truth tests, and provide more practical insights from the perspective of QPE applications. For rainfall rates below $100\ mm\ h^{-1}$, the $R(K_{\mathrm{DP}})$ agrees rather well with the gauge observations, and the selection of $K_{\mathrm{DP}}$ estimation method or controlling factor has minimal impacts on the QPE performance provided that the used controlling factor is not too extreme. For higher rain rates, significant underestimation is found for the

$R(K_{\mathrm{DP}})$, and a smaller window length results in higher $K_{\mathrm{DP}}$ thus less underestimation of rain rates. We show that the "best $K_{\mathrm{DP}}$ estimate"-based QPE cannot reproduce the gauge measurement of $201.9\ mm\ h^{-1}$ with commonly used assumptions for $R(K_{\mathrm{DP}})$, and potential responsible factors are discussed. We further show that the gauge with the $201.9\ mm\ h^{-1}$ report was at the vicinity of local rainfall hot spots during $16:00 \sim 17:00$ LST, while the 3-h rainfall accumulation center was located at the southwest of Zhengzhou city.

## 1 Introduction

Extreme rainfall can lead to high-impact events, such as soil erosion, debris flows and flash floods, and therefore poses a serious threat to both life and properties. In a warming climate, the occurrence frequency of regional extreme rainfall events is expected to increase (Allan and Soden, 2008; Donat et al., 2016), and this increase is particularly highlighted in regions of rapid urbanization (Zhang, 2020) where both the intensity of precipitation and the risk of flooding tend to be exacerbated

(Zhang et al., 2018).

To mitigate potential damages induced by extreme rainfall events, great efforts have been devoted to improving the prediction and monitoring of extreme rainfall. While the prediction technologies based on numerical models are confronting major

challenges (Luo et al., 2020), a collection of in-situ and remote sensing instruments is in operation to observe precipitation, thanks to the development of surface observing systems. The "ground truth" of surface precipitation map is customarily made from rain gauge observations. However, the rain gauge spacings are usually beyond several kilometers, and such "point" observations are inadequate to represent the localized rainfall centers produced by rapidly evolving storms (Schroeer et al., 2018). Gauge measurements seem to be falling short to support flood controlling in urban areas, where the inhomogeneity of underlying surfaces and complexity of fine-grained drainage connections call for rainfall observations with fine resolutions (Paz et al., 2020) and the simulated runoff is even more sensitive to the spatial resolution than to the temporal resolution (Bruni et al., 2015). The areal rainfall map can be seamlessly made with remote sensing observations. Weather radars have been used for quantitative precipitation estimation (QPE) based on equivalent radar reflectivity factor ($Z_e$), polarimetric observations (differential reflectivity $Z_{DR}$, specific differential phase $K_{DP}$, and cross correlation coefficient $\rho_{HV}$) or attenuation effects. From the perspective of raindrop size distribution (DSD) moments, $K_{DP}$ and specific attenuation, corresponding to the estimators of $R(K_{DP})$ and $R(A)$, respectively, are better correlated with rain rates. Therefore, $R(K_{DP})$ and $R(A)$ approaches are more efficient than $Z_e$-based ones in reducing uncertainties caused by the DSD variability (Ryzhkov et al., 2022). For lower rain rates, $R(A)$ has shown apparent advantages, whereas $R(K_{DP})$ is optimal for heavy rain (Ryzhkov et al., 2022). However, the accuracy of $K_{DP}$ estimation can significantly depend on the methods used (Reimel and Kumjian, 2021). To the best of our knowledge, the performance of $K_{DP}$-based heavy rainfall estimation has hardly been addressed despite a large volume of works on radar-based QPE (Schleiss et al., 2020; Cremonini et al., 2022).

On 20 July 2021, a devastating rainfall event hit Zhengzhou (Fig. 1a), one of the largest cities in central China, which hosts over 12 million residents. This event took place following the continuous, relatively weaker, rainfall on 18 and 19 July, and caused severe flooding over Zhengzhou city that led to around 300 fatalities and tremendous economic losses (Yin et al., 2022). In Zhengzhou city, urban infrastructure is mostly constructed with impervious materials, the so-called "gray urbanization" (gray area in Fig. 1b), making the city vulnerable to waterlogging in the presence of short-duration extreme rainfall. Given the limited emergency resources, it is therefore imperative to accurately locate the worst hit area. The most intense rainfall was produced during $14:00 \sim 17:00$ local solar time (LST) on July 20 (Yin et al., 2022) (Fig. 1c). Although a gauge (the site is marked with a black cross in Fig. 1a, b) located in Zhengzhou reported the maximum hourly rainfall of 201.9 mm at 17:00 LST, an hourly rainfall rate exceeding or close to the historical record in mainland China (Ding, 2019), location and extremity of other local rainfall hotspots are still unclear.

In this study, we aim to quantitatively assess the performance of different $K_{DP}$-estimation algorithms in this extreme rainfall event and analysis the areal precipitation map over Zhengzhou city. The paper is organized as follows. The data and $K_{DP}$ estimation methods are introduced in section 2. The methods of comparing $K_{DP}$ estimates from different algorithms, constructing different $R(K_{DP})$ estimators, and merging radar observations at multiple elevation angles are described in section 3. Section 4 compares the QPE performance of $K_{DP}$ estimated from different approaches. The areal precipitation map over Zhengzhou city is analyzed in section 5, and conclusions are given in section 6.

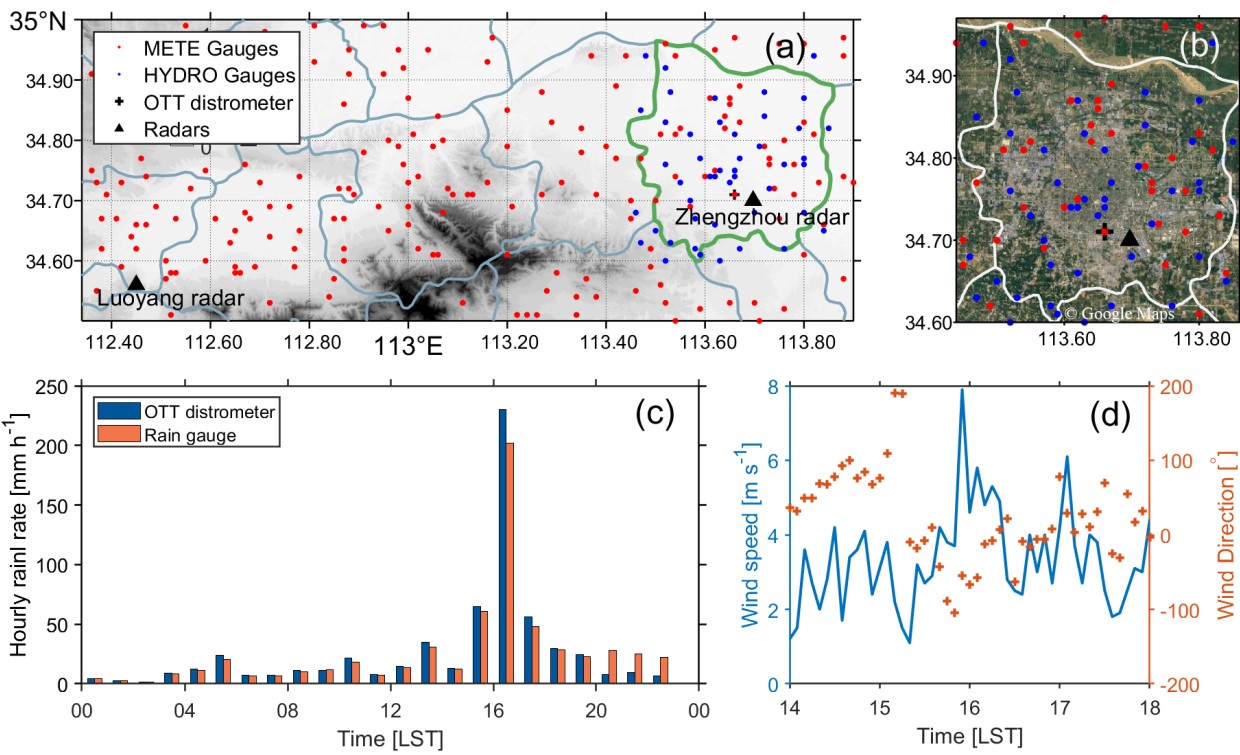

**Figure 1.** (a) Topography over and around Zhengzhou overlaid with the two operational S-band dual-polarization radars (black triangles), meteorological rain gauges (METE gauges; red dots), hydrological rain gauges (HYDRO gauges; blue dots) and one OTT disdrometer (black cross). (b) Satellite image of Zhengzhou city (modified from Google Maps). (c) Hourly rain rate recorded by the gauge and the OTT disdrometer located at the Zhengzhou national reference climatological station (113.66 °E, 34.71 °N, the site where the OTT disdrometer is deployed) on 20 July 2021. (d) 5-min horizontal wind speed (left) and direction (right) from 14:00 to 18:00 LST. The light blue curves in (a) indicate county boundaries and Zhengzhou city is outlined in dark green. Note that the HYDRO gauges are widely distributed, although only those over Zhengzhou city are presented in (a).

## 2  Data

### 2.1  Dual-polarization weather radars

Since the late 1990s, a nationwide weather radar network composing of over 200 China's New Generation Doppler Weather Radars (CINRADs) has been built in China. CINRADs typically work in the volume coverage pattern 21 mode, which consists of nine plan position indicator scans (0.5° , 1.5° , 2.4° , 3.3° , 4.3° , 6.0° , 9.9° , 14.6° , and 19.5°) with the volumetric update time of 6 min. In recent years, more than 100 CINRADs have been upgraded to dual-polarization systems and others are in progress. As shown in Fig. 1a, two S-band dual-polarization CINRADs are deployed in Luoyang city (112.44 ° E, 34.5 ° N) and Zhengzhou city (113.697 ° E, 34.704 ° N), respectively. Both Luoyang and Zhengzhou radars have the same configurations,

60

e.g., the range resolution of 0.25 km, azimuth resolution of 1 °, and the time resolution of 6 min. Mt. Song, located between Luoyang and Zhengzhou, is about 0.9 km above mean sea level (amsl), and the altitude of Luoyang radar is 0.209 km amsl. Therefore, the mountains partially block Luoyang radar's lowest radar beam (0.5°), which may affect reflectivity observations but $K_{\mathrm{DP}}$ is immune to this effect (Lang et al., 2009). The altitude of Zhengzhou radar is 0.18 km. We have checked Luoyang and Zhengzhou radar observations at different elevation angles, and no second-trip echoes can be identified. Due to the power outage, the Zhengzhou radar data were missing from 17:18 to 19:48 LST. Still, this extreme precipitation event over Zhengzhou city was successfully captured by the Zhengzhou radar, since the majority of the precipitation system moved out of urban Zhengzhou after 17:00 LST.

$K_{\mathrm{DP}}$ is one-half the range derivative of differential phase shift ($\Phi_{\mathrm{DP}}$), while radars measure the total differential phase shift which is a combination of $K_{\mathrm{DP}}$ and backscatter differential phase ($\delta$). The impact of $\delta$ on $K_{\mathrm{DP}}$ is negligible at S-band, while it can be significant at shorter radar wavelengths (Trömel et al., 2013). There are a number of algorithms available for $K_{\mathrm{DP}}$ estimation, and some of them are accessible in the open-source tool Py-ART (Helmus and Collis, 2016). Reimel and Kumjian (2021) used a known-truth framework to evaluate the commonly used $K_{\mathrm{DP}}$ estimation algorithms. They found that the algorithm accuracy is dependent on the raw $\Phi_{\mathrm{DP}}$, and concluded that each algorithm has its apparent strengths and weakness. They also showed that the method of Maesaka et al. (2012) and linear programming (Giangrande et al., 2013) can change the overall behavior between oversmoothing and undersmoothing. This means that a couple of $K_{\mathrm{DP}}$ estimates generated with different tuned parameters may yield a range of values where the "best $K_{\mathrm{DP}}$" falls in, despite that it is challenging to determine the best controlling parameter. In this study, we will assess the performance of using different tuning parameters in $K_{\mathrm{DP}}$-based QPE. A brief introduction of $K_{\mathrm{DP}}$-estimation algorithms is given below.

– The operationally used $K_{\mathrm{DP}}$ estimation algorithm in CINRADs is a traditional least square fitting (LSF). As a regression approach, LSF is easy to implement and is commonly used for estimating $K_{\mathrm{DP}}$ in weather radars. For a given window of smoothed $\Phi_{\mathrm{DP}}$, linear regression is done to estimate $K_{\mathrm{DP}}$. The window length is adaptive and depends on observed $Z_{\mathrm{e}}$ (Wang and Chandrasekar, 2009). Due to this dependence on $Z_{\mathrm{e}}$, which can be affected by data quality issues such as ground clutter, $K_{\mathrm{DP}}$ estimates with $\rho_{\mathrm{HV}}$ below 0.8 are removed.

– Linear programming (LP). This algorithm assumes that $\Phi_{\mathrm{DP}}$ monotonically increases with range and uses self-consistency between $Z_{\mathrm{e}}$ and $K_{\mathrm{DP}}$. Since the self-consistency relation is developed for rainfall, the algorithm does not process $\Phi_{\mathrm{DP}}$ values above melting layer (4.5 km in this study) or in presence of hail. The algorithm is proposed by Giangrande et al. (2013) and is compiled in Py-ART (Helmus and Collis, 2016). The user can define a self-consistency coefficient for $K_{\mathrm{DP}}$-$Z_{\mathrm{e}}$ as well as a self-consistency factor or use the default settings. In Py-ART, the self-consistency factor is used to define the weight of the $Z_{\mathrm{e}}$-$K_{\mathrm{DP}}$ relationship on the final solution, and the default value is $6\times10^4$. For S-band radars, the self-consistency factor below $4\times10^4$ may degrade the estimation performance (Reimel and Kumjian, 2021), while it should be tuned at C-band (Cremonini et al., 2022). In this study, the default setting in Py-ART was used. We have compared the $\Phi_{\mathrm{DP}}$ reconstructed by the LP method with the raw $\Phi_{\mathrm{DP}}$ in radar radials, and found that the algorithm works reasonably well. In addition, the user should set a window length in which a Sobel filter is imposed, and the length of

this window effectively affects the smoothness of the $K_{DP}$ field. For a comparison with (Reimel and Kumjian, 2021), we have tried the window lengths of 5 (0.75 km), 15 (3.75 km), 25 (6.25 km), 35 (8.75 km) and 45 (11.25 km) in this study.

– Maesaka algorithm. This algorithm assumes monotonic increase of $\Phi_{DP}$ below the melting layer, namely applicable in rain. It applies a low-pass filter to smooth the observed $\Phi_{DP}$, and the effect that the low pass filter has on the final solution depends on a user-defined parameter Clpf. By changing the value of Clpf the user can control the amount of smoothing applied by the algorithm. A thorough introduction of the algorithm is referred to (Maesaka et al., 2012). Similar to Reimel and Kumjian (2021), we have used values of $10^0, 10^2, 10^4$ and $10^6$ for Clpf in this study for $K_{DP}$ estimation.

Note that the data quality of $\Phi_{DP}$, which is also critical for $K_{DP}$ estimation, can be heavily affected by ground clutter which usually leads to significant spikes of $\Phi_{DP}$ at certain ranges ($r$). To minimize the impact of those spikes on $K_{DP}$ estimation, the following procedures were utilized:

     – Firstly, a linear fit was made to the raw $\Phi_{DP}(r)$ data for an interval of 5 km. The fitted values were labeled as $\Phi'_{DP}(r)$.

     – Then, the point with $|\Phi_{DP}(r)\text{-}\Phi'_{DP}(r)|$ exceeding $10°$ was identified as clutter.

– Finally, a cubic spline interpolation was made to the identified clutter points.

These steps can effectively remove majority of clutter signals, however, local perturbation of $\Phi_{DP}$ can be on the order of $10°$ given the area of interest is so close to the radar. Therefore, we have also manually checked the $\Phi_{DP}$ fields and removed significant clutter signals.

## 2.2   Surface observations

The most widely used rainfall measuring instrument in operational weather services is the tipping bucket rain gauge. The buckets are mounted on a fulcrum and located below a funnel. Once one bucket is filled with water channeled through the funnel, it tips down and the other bucket raises. At the same time, a switch records an electronic signal, which is then converted to the amount of rain. The gauge observations used in this study are from both meteorological (METE) and hydrological (HYDRO) rain gauge stations, respectively. For the METE gauges, the volume of a bucket is 0.1 mm, which corresponds

to the minimal detectable rain accumulation of 0.1 mm. Every one minute, the number of tips is recorded. Liu et al. (2019) have pointed out that the uncertainty of such gauges is about 4 % for rain rates exceeding 10 mm h$^{-1}$. The HYDRO gauges employ tipping buckets as well, but the instrument model differs from that of the METE gauges. The minimal detectable rain accumulation of the HYDRO gauges is 0.5 mm and the time resolution is 1 h. The high temporal resolution of the METE gauges enables the inspection of the data quality. For the HYDRO gauges with hourly measurements, the inverse distance

weighting (IDW) approach (Chen and Liu, 2012) was implemented to yield an estimate of hourly rainfall accumulation at a given HYDRO gauge site. Then, the observed value below 50% of the expected one was removed, in order to indentify the gauges which were not working due to power outages. After the data quality control, 114 gauges were used in this study.

Different from tipping buckets gauges, OTT PARSIVEL disdrometer (OTT) measures rainfall by accounting every raindrop that severely attenuates the light signal emitted from a laser sheet. This different measuring principle makes the OTT an independent instrument that can be used to evaluate gauge observations. The one deployed close to the gauge is the second generation of OTT. Figure 1 (c) compares hourly rain rate measurements recorded by a rain gauge and the OTT at Zhengzhou national reference climatological station in 20 July 2021. During most of the period, OTT slightly overestimates hourly rainfall accumulations compared to the gauge observations. This may be attributed to the overestimation of large drops possibly caused by several factors, such as the assumed oblate shape and the coincidence effect (Tokay et al., 2013; Park et al., 2017).

## 2.3 Comparison of Luoyang and Zhengzhou radar observations

Zhengzhou radar is located in the southeast of Zhengzhou city and Luoyang radar is around 120 km away from the Zhengzhou city. Since the lowest beam of Luoyang radar is about 2.2 km over the Zhengzhou city while the lowest beams of Zhengzhou radar are rather close to the surface, the agreement between Luoyang and Zhengzhou radar observations is an potential issue that should be addressed. Given the hourly precipitation from 16:00 to 17:00 LST reached the peak, radar retrievals during this period were used for an assessment. To provide a reference for the operational service, we used $K_{DP}$ from CINRAD's operational products (LSF method) in the comparison. The lowest elevation angle of Luoyang radar ($0.5°$, the radar beam is about 2.2 km over Zhengzhou city) was used, while the selection of $1.5°$ for the Zhengzhou radar was due to significant clutter issues at $0.5°$. A linear interpolation was applied to range gates that were severely affected by ground clutter as characterized by $\rho_{HV}$ below 0.8. The raw data were interpolated into the spatial resolution of 0.5 km using Py-ART (Helmus and Collis, 2016). Note that we did not find significant evidence of hail from Luoyang radar $\rho_{HV}$ observations, and therefore hail is anticipated to be absent below 2.2 km.

As shown in Fig. 2, the heaviest rainfall poured over the area around the Zhengzhou radar site during $16:00 \sim 17:00$ LST, which may explain the breakdown of Zhengzhou radar at 17:12 LST. A closer inspection to Fig. 2b shows that the location of precipitation center retrieved from the Luoyang radar (black isolines) is on the east side of that from the Zhengzhou radar. Yin et al. (2022) have made numerical simulations of this event, and they found that the storms were vertically tilted eastward. The sampling volume of the Luoyang radar over Zhengzhou city was about 2 km, while the Zhengzhou radar observed near-surface precipitation. Therefore, the precipitation observed by the Luoyang radar is more eastward than the Zhengzhou radar. In addition, warm rain processes may also significantly augment rain rates within the height of 2 km (Yu et al., 2022). Given the effects discussed above, Zhengzhou radar observations were used for QPE in this study.

## 3 Methods

As pointed out by Bringi and Chandrasekar (2001), the accuracy of $K_{DP}$-based QPE is dependent on not only the $K_{DP}$ estimation from radars but also on the parameterization of $R(K_{DP})$. This section will address these two aspects respectively.

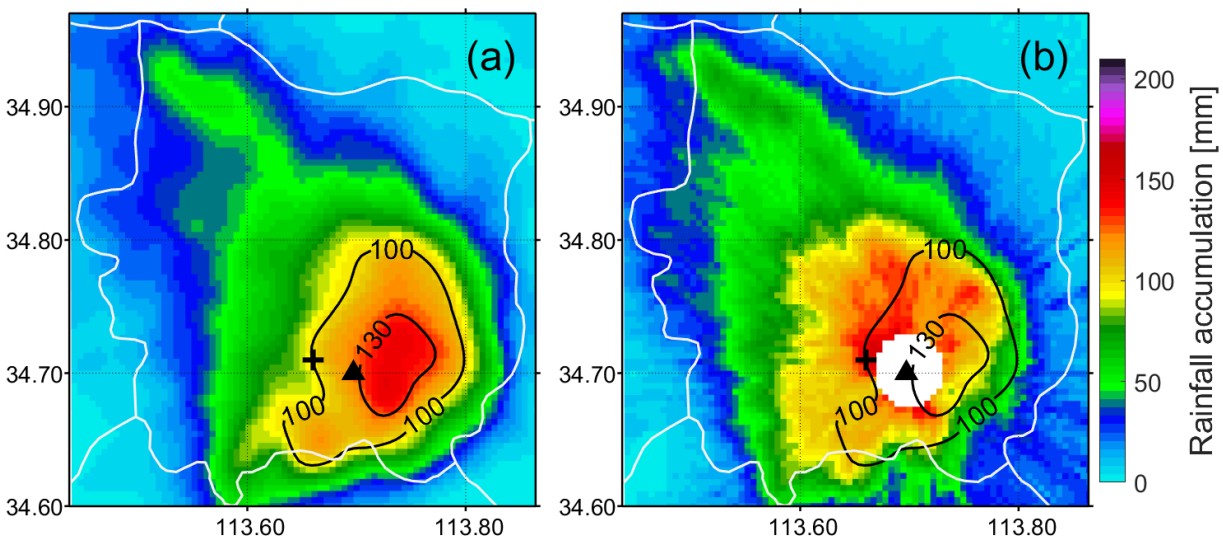

**Figure 2.** Rainfall accumulation from 16:00 to 17:00 LST estimated using $R = 51 K_{\mathrm{DP}}^{0.86}$, in which $K_{\mathrm{DP}}$ estimates were from the operational data products (LSF method). (a) Luoyang radar data at the elevation angle of $0.5°$ and (b) Zhengzhou radar data at the elevation angle of $1.5°$ were used for comparison. Note that $K_{\mathrm{DP}}$ estimates within 3 km to the Zhengzhou radar site were removed. The black triangle and cross denote the Zhengzhou radar and the gauge/OTT site, respectively. The black isolines indicate the rainfall accumulation of 100 mm and 130 mm observed by the Luoyang radar, respectively.

## 3.1 Approaching the "best $K_{\mathrm{DP}}$ estimate"

The calculation of $K_{\mathrm{DP}}$ with Maesaka and LP algorithms requires a presetting of Clpf and window length, respectively, which controls the extent of smoothing applied to $\Phi_{\mathrm{DP}}$. Bringi and Chandrasekar (2001) concluded that the minimal window length required for $K_{\mathrm{DP}}$ estimation decreases with precipitation intensity. Reimel and Kumjian (2021) further showed that the "best $K_{\mathrm{DP}}$ estimate" falls in a range of values produced by varying the parameters in known-truth simulations and the retrieved $K_{\mathrm{DP}}$ is heavily dependent on the algorithm and tuning parameter employed for steep real $K_{\mathrm{DP}}$ regions. In this study, the Zhengzhou national reference climatological station hosts an OTT and the gauge with the 201.9 mm h$^{-1}$ report and is 3.15 km at 274° azimuth of Zhengzhou radar site. $K_{\mathrm{DP}}$ estimates from different algorithms with various tuning parameters over this site were compared. Here, radar observations at the elevation angles of 1.5°, 2.4°, 3.3° and 4.3° were used for the following considerations. (1) The dependence of observed $K_{\mathrm{DP}}$ on the viewing angle is expected to be negligible at small radar elevation angles, i.e., smaller than 4.3° (Bringi and Chandrasekar, 2001). (2) Due to the strong ground clutter contamination, we discarded the data recorded at the lowest elevation angle. $K_{\mathrm{DP}}$ estimates at elevation angles of 1.5°, 2.4°, 3.3° and 4.3° corresponding to heights about 0.083 km, 0.132 km, 0.182 km and 0.237 km, respectively, over the station were used. Given the small range of height, we assume that the real $K_{\mathrm{DP}}$ values over the Zhengzhou station at these elevation angles were about the same.

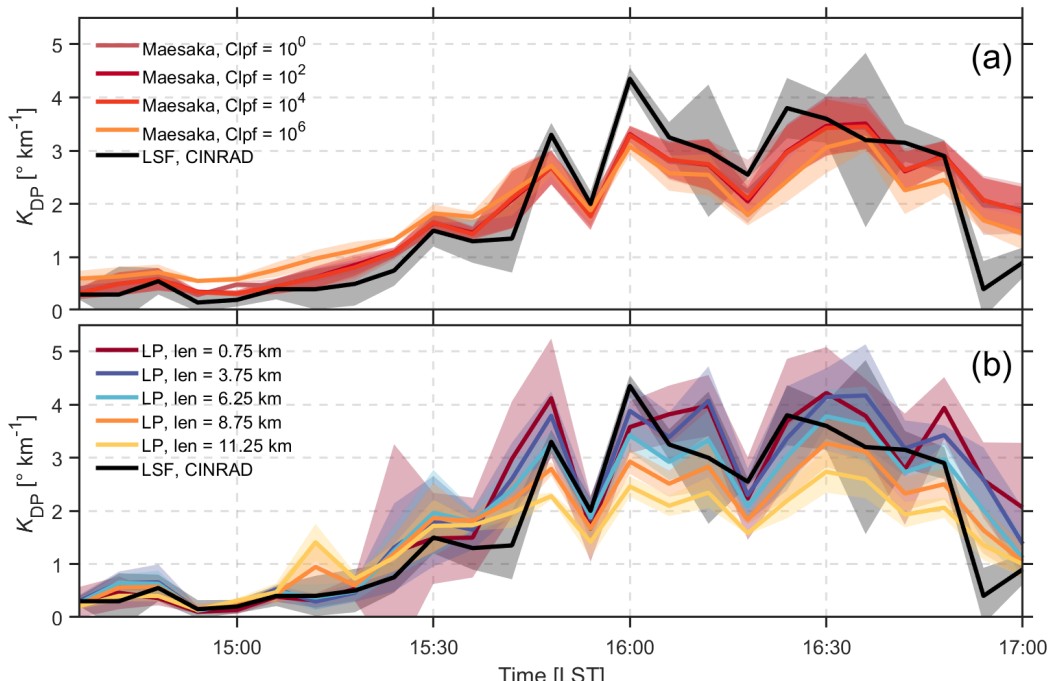

**Figure 3.** $K_{DP}$ estimates using (a) Maesaka (2012) method and (b) LP over Zhengzhou national reference climatological station. Thick lines and shading areas indicate the median values and standard deviations of $K_{DP}$ at elevation angles of $1.5°$, $2.4°$, $3.3°$ and $4.3°$. LP: linear programming method (Giangrande et al., 2013); LSF: least square fitting, the CINRAD's operational algorithm. Colored lines indicate different window length (len) used in LP.

Bearing the considerations above, $K_{DP}$ estimates using Maesaka and LP algorithms are presented in Fig. 3. Interestingly, our results resemble what is presented in Fig. 16 of (Reimel and Kumjian, 2021) in following aspects:

– Stronger dependence of $K_{DP}$ on the tuning parameter is found for LP than the Maesaka algorithm.

– Smaller window length used in the LP method generally leads to higher $K_{DP}$ in heavy rainfall periods. In comparison, $K_{DP}$ does not significantly change by varying Clpf from $10^0$ to $10^4$ for the Maesaka algorithm.

– LP can produce higher $K_{DP}$ values than the Maesaka algorithm.

– In presence of relatively light rainfall, such as before 15:00 LST, longer window length in LP agrees better with the Maesaka algorithm.

– $K_{DP}$ values retrieved from both the LSF and Maesaka algorithms are less uncertain than LP.

However, the impact of changing the window length does not seem to be as significant as in (Reimel and Kumjian, 2021). The $K_{DP}$ values with a window length of 0.75 km which is expected to yield nearly the most extreme $K_{DP}$ (Reimel and

Kumjian, 2021) are comparable with the window length of 3.75 km (Fig. 3b). Namely, it appears that the $K_{DP}$ estimated from the LP algorithm has reached "saturation" at the window length of 3.75 km.

It should be noted that the non-uniform radar beam filling was not considered in idealized known-truth tests (Reimel and Kumjian, 2021), but it can lead to local perturbation of $K_{DP}$ (Ryzhkov and Zrnic, 1998). As the LP and Maesaka algorithms assume the monotonic increase of $\Phi_{DP}$, they are expected to yield higher $K_{DP}$ than the LSF method if the negative radial slope of $\Phi_{DP}$ occurs in the close proximity. However, this effect does not seem to be significant in this study for the following reasons. Firstly, the Zhengzhou radar is close to the gauge site (3.15 km), and therefore the radar sampling volume is much smaller than that at larger distances. Then, the gauge site was not located in the edges of rain cells (see merged $K_{DP}$ observations at https://github.com/HaoranLiHelsinki/Figs_Zhengzhou). Finally, we have manually checked $\Phi_{DP}$ observations, and did not see significant negative radial slope of $\Phi_{DP}$. In addition, the smallest Clpf (least smoothing) yields smaller $K_{DP}$ than the LSF method from 16:00 to 17:00 LST (Fig. 3a), suggesting the selection of $K_{DP}$ estimation method is more important than the effect of non-uniform radar beam filling in this study.

## 3.2 Parameterizations of $R(K_{DP})$

While $K_{DP}$ is less dependent on DSDs than other radar products, a localized $R(K_{DP})$ parameterization is suggested to minimize the impact of varying DSDs (e.g., Chen et al., 2022). In this study, the OTT disdrometer observations on 20 July 2021 were used as input to PyTMatrix (Leinonen, 2014) to calculate radar polarimetric variables. Before the calculation, we have removed raindrops with the velocity outside of $\pm 50\%$ of empirical relations (Atlas et al., 1973) or with the volume equivalent diameter higher than 6 mm. It was assumed that raindrops are oblate spheroids with the aspect ratio parameterized by the equivolumetric spherical drop diameter (Thurai et al., 2007). The water temperature was set to 20 °C, and the orientation of rain drops was assumed to be normally distributed with zero mean and a certain value of standard deviation ($\sigma$). We will discuss the factors affecting the accuracy of $R(K_{DP})$ parameterization as follows.

– DSDs. Zhang et al. (2022) have shown that for a given $K_{DP}$ the fitted relation for OTT observations during $16:00 \sim 17:00$ LST yields higher precipitation rates than that for the whole day, but the value does not exceed $\sim 15$ mm h$^{-1}$. In addition, most rain rates above 200 mm h$^{-1}$ are during $16:00 \sim 17:00$ LST, and they follow the fitted curves rather well. Therefore, we have used the OTT data from 00:00 to 24:00 LST 20 July 2021.

– Assumed $\sigma$. The simulated radar polarimetric variables are dependent on $\sigma$ if hydrometeors are assumed to be spheroids (Li et al., 2018). Bringi et al. (2008) have found a $\sigma$ of around 7° in a stratiform rainfall event with low wind conditions and 12° in moderate wind conditions. In presence of high winds, this value can be 13.6° $\sim$ 24.7° (Bolek and Testik, 2022). The automatic weather station at the OTT site reported that wind speed during this event ranged from 2 to 5 $m\ s^{-1}$ with a peak of 7.8 $m\ s^{-1}$ at around 16:00 LST. The magnitude of wind speed seems rather close to the condition corresponding to the $\sigma$ of 13.6° (Bolek and Testik, 2022).

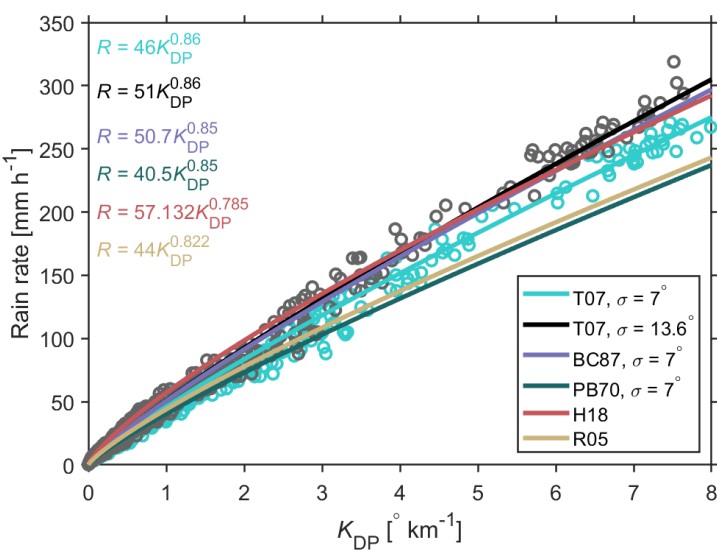

**Figure 4.** T-Matrix-based simulation of $K_{\mathrm{DP}}$ versus rain rate from the OTT observations on 21 July 2021. Black and green circles indicate observations with the $\sigma = 7$ and $13.6^\circ$, respectively, assuming the aspect ratio parameterization from (Thurai et al., 2007, T07). The $R(K_{\mathrm{DP}})$ relations from (Ryzhkov et al., 2005, R05), (Huang et al., 2018, H18) as well as (Bringi and Chandrasekar, 2001) with aspect ratio parameterizations from (Pruppacher and Beard, 1970, PB70) and (Beard and Chuang, 1987, BC87) are also presented.

For a given $K_{\mathrm{DP}}$ of $5\,^\circ\,km^{-1}$, the estimated rain rates are $203.6\ mm\ h^{-1}$ and $183.6\ mm\ h^{-1}$ for $\sigma$ of $13.6^\circ$ and $7^\circ$,
respectively. This value can even be $279.4\ mm\ h^{-1}$ ($R = 70K_{\mathrm{DP}}^{0.86}$, not shown) for a $\sigma$ of $24.7^\circ$, which was observed in a tornadic squall-line storm (Bolek and Testik, 2022) and seems to be unrealistically large in this case.

– Aspect ratio parameterization. Assuming a light wind condition ($\sigma = 7^\circ$), the (Pruppacher and Beard, 1970) and (Beard and Chuang, 1987) parameterizations lead to quite different rain rate estimations (Fig. 4), as earlier shown by Bringi and Chandrasekar (2001). Thurai et al. (2007) have shown that the observed raindrop shapes are rather close to the model
simulations in (Beard and Chuang, 1987). This is the reason why we have employed the (Thurai et al., 2007) aspect ratio parameterization in the $K_{\mathrm{DP}}$ calculations.

As shown in Fig. 4, the deviation between different parameterizations seems relatively small for smaller rain rates, but significantly enlarges as the precipitation intensity increases. This indicates that a single $R(K_{\mathrm{DP}})$ parameterization is applicable for QPE of moderate rainfall. For higher rain rates, the fitted relation for $\sigma$ of $13.6^\circ$ agrees rather well with (Beard and Chuang,
1987) and (Huang et al., 2018).

### 3.3 Merge of Zhengzhou radar observations at multiple elevation angles

One of the major challenges of using weather radar observations is to mitigate the ground clutter contamination in the vicinity of radar sites. To remove pixels affected by ground clutters, the threshold of $\rho_{\rm hv} = 0.8$ (Kumjian, 2013) was implemented firstly. In the second step, with the assumption that the rain microphysics within 0.6 km to the surface do not change, the median of radar observations at elevation angles from $0.5°$ to $6.0°$ was used to replace the pixels identified as ground clutter. Because of the rapid increase of the beam height at higher elevation angles, the maximum radar range decreases with the increase of elevation angle for a given height. Due to the clutter contamination, very few radar observations at the vicinity of the radar site at the elevation angle of $0.5°$ were used in the data merge. Meanwhile, radar data at $9.9°$, $14.6°$, and $19.5°$ were discarded given limited valid data and the elevation dependence of polarimetric measurements may start appearing (Bringi and Chandrasekar, 2001). Then, the Inverse Distance Weighting (IDW) interpolation (Cressman, 1959; Goudenhoofdt and Delobbe, 2009) of the radar data was applied to filling in empty regions, and the new constructed radar data were interpolated into the spatial resolution of 0.5 km using Py-ART (Helmus and Collis, 2016).

## 4 Results

### 4.1 $K_{\rm DP}$-based QPE over the gauge/OTT site

With a parameterized $R(K_{\rm DP})$, we have been able to quantitatively analyze the performance of $K_{\rm DP}$-based QPE over the gauge site. Given the high rain rates in this event, $K_{\rm DP}$ estimates using the LSF method, the Maesaka algorithm with Clpf = $10^0$ as well as the LP method with the window length of 0.75 km are used for comparison. As shown in Fig. 5(a) and (b), $R(K_{\rm DP})$ agrees generally well with the gauge and OTT observations before 16:00 LST, regardless of the $K_{\rm DP}$ estimation method or the used $R(K_{\rm DP})$ parameterizations.

From 16:00 to 17:00 LST, significant deviations can be found between the gauge and OTT observations. In addition, $K_{\rm DP}$-based QPE significantly underestimates the surface precipitation during this period. With a larger $\sigma$ (Fig. 5b), $R(K_{\rm DP})$ is still well below OTT/gauge observations. Therefore, it is of necessity to discuss factors potentially contributing to this underestimation.

– Accuracy of $K_{\rm DP}$ estimates. Compared with the LSF and Maesaka algorithms, $K_{\rm DP}$ estimated by the LP method less underestimates the rainfall. Note that the parameterizations used for Maesaka and LP algorithms are expected to generate the highest $K_{\rm DP}$ values in heavy rainfall (Reimel and Kumjian, 2021). Therefore, we should have good confidence that the best $K_{\rm DP}$ should be close to or lower than the estimates.

– DSD variations in the air. The lowest radar sampling volume is 0.083 km over the gauge/OTT site $(1.5°)$ while the highest is 0.237 km $(4.3°)$. If DSDs would have significantly varied, $K_{\rm DP}$ estimates at different elevation angles should also change. However, the uncertainty of $K_{\rm DP}$ estimates at different elevations angles is on the order of $0.5\ °km^{-1}$. Therefore, the change of DSDs should not be significant, and the DSDs observed by OTT should be applicable to radar

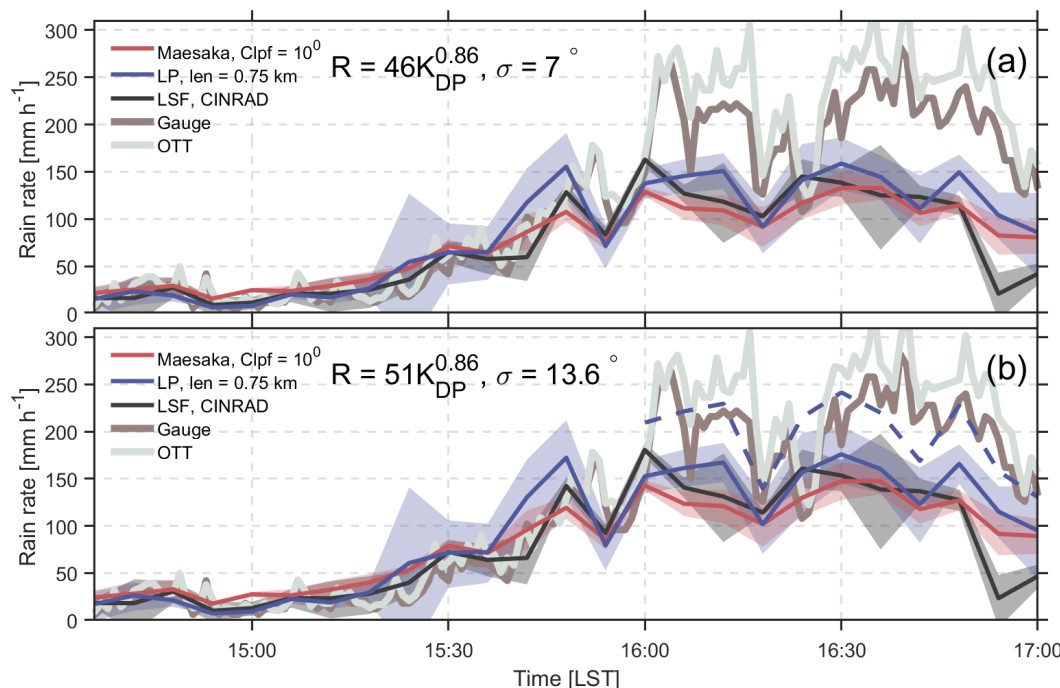

**Figure 5.** Comparison of rainfall estimates using $K_{DP}$ estimated from different methods over Zhengzhou national reference climatological station. Thick lines and shading areas indicate median values and standard deviations of rain rates estimated from $K_{DP}$ at elevation angles of $1.5°$, $2.4°$, $3.3°$ and $4.3°$. The used parameterizations are (a) $R = 46K_{DP}^{0.86}$ and (b) $R = 51K_{DP}^{0.86}$, respectively. The dashed line in (b) is the use of $R = 70K_{DP}^{0.86}$ ($\sigma = 24.7°$) for QPE from 16:00 to 17:00 LST.

observations that are so close to the surface. In addition, the rain water content does not seem to change within such a short distance (Chen et al., 2020).

– Vertical air motions. The $K_{DP}$-based QPE assumes the absence of vertical air motions. For a given DSD in the radar sampling volume, downdrafts can lead to the underestimation of rain rates. For such heavy rainfall, a downdraft of $2 \sim 3$ $m\,s^{-1}$ can lead to the rain rate underestimation of $30 \sim 40\,\%$. We have examined this factor from two aspects. Firstly, we found that the diameter-velocity diagram generated by OTT observations agrees rather well with the empirical relation, suggesting the absence of significant downdrafts near the surface (Kim and Song, 2018).

Then, although direct retrieval of vertical air motions is rather uncertain (Oue et al., 2019) compared with the magnitude of expected downdrafts of $1 \sim 2\,m\,s^{-1}$ as shown in model simulations (Yin et al., 2022), existence of downdrafts is detectable in radial divergence (Roberts and Wilson, 1989; Adachi et al., 2016). Here, we define the radial divergence (RD) as

$$RD = \frac{V_{i+4} - V_{i-4}}{r_{i+4} - r_{i-4}} \qquad (1)$$

where $V_i$ is the observed radar Doppler velocity at the range gate $r_i$. The RD is derived every 2 km for a range resolution of 0.25 km according to Eq. 1. Figure 6 shows time series of the observed Doppler velocity (black) as well as RD (red) over Zhengzhou national reference climatological station. The leading edge of the extreme-rainfall-producing storms passed the site at about 15:36 LST when the Doppler velocity underwent the transition from positive to negative and the RD reached a local minimum ($-3 \times 10^{-3} \ s^{-1}$), indicating the presence of updrafts. From 16:00 to 17:00 LST, the Doppler velocity is around $0 \ m \ s^{-1}$ and RD is about $2 \times 10^{-3} \ s^{-1}$, suggesting the sustained downdrafts. Therefore, the unquantified downward air motions may be responsible for the underestimation of rainfall accumulation.

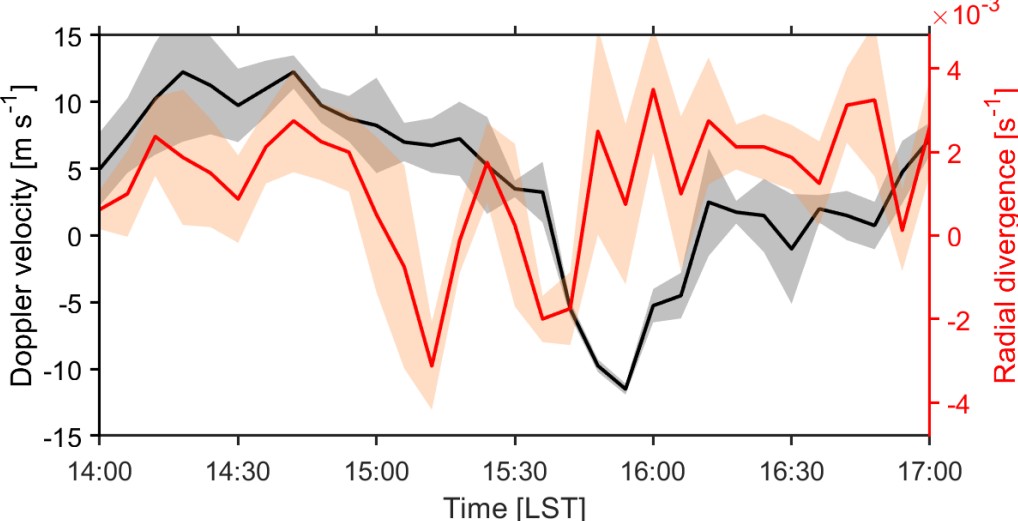

**Figure 6.** Doppler velocity (left) and Radial divergence (right) observed over Zhengzhou national reference climatological station. Thick lines and shading areas indicate the median values and standard deviations at elevation angles of $1.5°$, $2.4°$, $3.3°$ and $4.3°$.

– Assumption of $\sigma$. As shown in Fig. 4, the assumption on $\sigma$ is critical for the parameterization of $R(K_{DP})$. However, $\sigma$ cannot be measured by OTT, and very few experiments have been conducted for addressing this (e.g., Bringi et al., 2008; Bolek and Testik, 2022). The wind observations are rather close to what was reported by Bolek and Testik (2022), and $\sigma = 13.6°$ seems to be a good first guess. If the $\sigma = 24.7°$ measured during the passage of a tornadic squall-line storm (the 4-min running averaged horizontal wind speed is $6 \sim 10 \ m \ s^{-1}$) is used, the resulted rain rate estimation is rather close to gauge/OTT measurements (dashed line in Fig. 5b). However, the observed horizontal wind speed is $3 \sim 5 \ m \ s^{-1}$ from 16:00 to 17:00 LST (Fig. 1d). Therefore, even though we cannot give a more accurate estimate of $\sigma$, $24.7°$ seems to to be unrealistically large in this study.

- Different sampling volumes between the radar and the gauge/OTT. The width of the sampling volume for Zhengzhou radar with a beam width of $1°$ over the gauge site is about 55 m, which is much larger than that of a gauge. Although this effect is difficult to quantify, we argue that it plays a minor role for the rainfall underestimation. By manually checking the movement of storms (merged $K_{DP}$ observations at https://github.com/HaoranLiHelsinki/Figs_Zhengzhou), we found that the storm propagation speed is on the order of several kilometers per hour, contrasting with the much smaller radar sampling volume. Given the rapid changing nature of the storms, the sampling effect does not seem to be a major factor responsible for the rainfall underestimation.

## 4.2 Statistical evaluation

The dense meteorological and hydrological rain gauge network in Zhengzhou city allows a statistical evaluation of the $K_{DP}$-based QPE. In addition, $R(K_{DP})$ is expected to be less uncertain than other approaches in heavy precipitation (Ryzhkov et al., 2022). Therefore, the performance of $R(K_{DP})$ during the most intensive precipitation period ($14{:}00 \sim 17{:}00$ LST) was investigated. As discussed above, the assumption of $\sigma = 13.6°$ appears to be more suitable than the commonly used $7°$ in this event, thus $R = 51K_{DP}^{0.86}$ was used. Note that the gridded $R(K_{DP})$, as introduced in Sect. 3.3, was used for comparison.

For rainfall rates below 50 $mm\ h^{-1}$, the standard deviation ($std$) and bias of $R(K_{DP})$ are mostly on the order of $7 \sim 8$ $mm\ h^{-1}$ and $-1 \sim 0\ mm\ h^{-1}$, respectively. Regarding the LP method, the used window length does not significantly degrade the accuracy of QPE (Fig. 7a-e). The performance of the Maesaka method is comparable with that of the LP method (Fig. 7f-h), except for Clpf = $10^6$ (Fig. 7i) which imposes a too aggressive filter that obviously leads to oversmoothing as well as much larger std and bias. The operationally used LSF method (Fig. 7j) shows relatively large bias (1.8 $mm\ h^{-1}$), indicating that the $K_{DP}$ as derived from the LSF method in rainfall rates below 50 $mm\ h^{-1}$ should be used with caution.

For rainfall rates above 50 $mm\ h^{-1}$, $R(K_{DP})$ in general underestimates hourly rainfall accumulation, and this underestimation becomes more significant as the rain rate increases (smaller bias and std of red dots than those of black dots). $K_{DP}$ values estimated from the Maesaka algorithms is on average smaller than that from the LP and LSF methods, which is consistent with the results in Fig. 3. Interestingly, the std and bias of LP method are very close to those of the LST method regardless of the used window length. This indicates that varying the window length from 0.75 to 11.25 km has minimal impact on the accuracy of $R(K_{DP})$ for rain rates of $50 \sim 100\ mm\ h^{-1}$ in this event.

Reimel and Kumjian (2021) have shown that smaller window length employed in the LP method yields higher $K_{DP}$. This appears to be true for the gauge with the 201.9 $mm\ h^{-1}$ report, but decreasing the window length did not significantly ameliorate the underestimation in a statistical perspective (Fig. 7a-e). Specifically, the highest hourly rainfall accumulation was found for the LP method, and the value rises from 100 $mm\ h^{-1}$ (len = 11.25 km) to 149.6 $mm\ h^{-1}$ (len = 0.75 km). For a reference, the value was 122.9 $mm\ h^{-1}$ and 143.3 $mm\ h^{-1}$ for Maesaka method with Clpf = $10^0$ and LSF method, respectively.

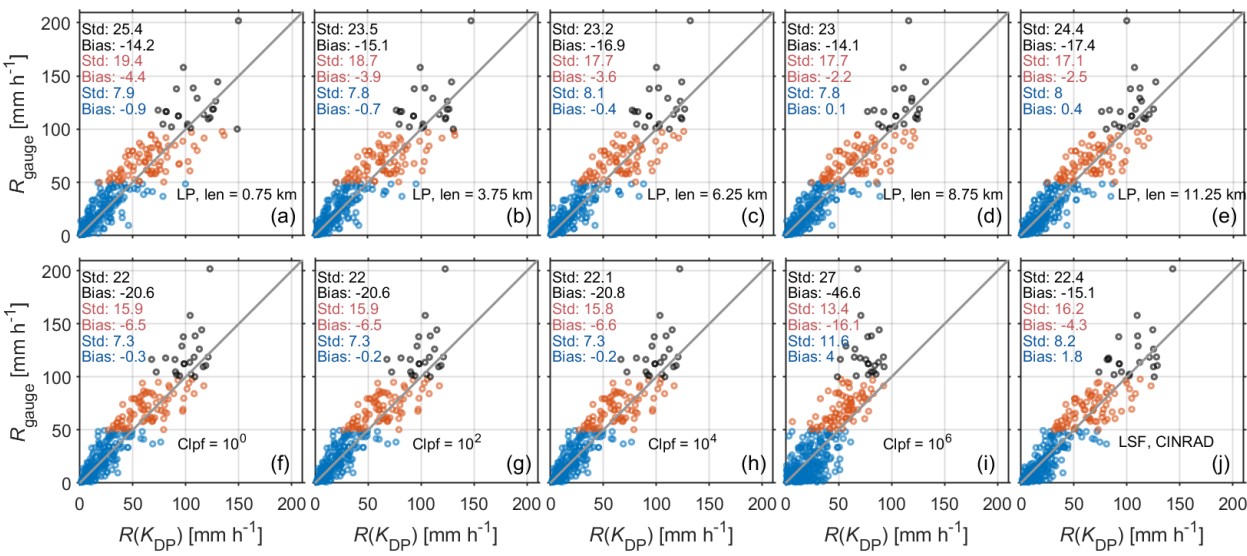

**Figure 7.** $K_{DP}$-based hourly rainfall accumulation v.s. gauge observations from 14:00 to 17:00 LST. $K_{DP}$ was estimated using (a-e) LP, (f-i) Maesaka and (j) LSF methods. Rain rates were divided into three groups: $R_{gauge} < 50\ mm\ h^{-1}$ (blue), $50\ mm\ h^{-1} \leq R_{gauge} < 100\ mm\ h^{-1}$ (red), and $100\ mm\ h^{-1} \leq R_{gauge}$ (black). The standard deviation (std) and bias between $R_{gauge}$ and $R(K_{DP})$ for each group are marked by corresponding colors. $R = 51K_{DP}^{0.86}$ was used.

## 5 Analysis of areal rainfall map

As discussed above, the use of window length (LP method) and Clpf (Maesaka algorithm) has limited impact on heavy rainfall QPE and the window length of 0.75 km generates the closest rainfall estimation to the 201.9 $mm\ h^{-1}$ report. Therefore, we have compared the areal hourly rainfall accumulation based on $K_{DP}$ generated by these three methods during the period with most intensive rainfall (14:00 $\sim$ 17:00 LST).

As shown in Fig. 8, the hot spots of rainfall rates can be manually identified and the results of the three methods generally agree with each other for $R(K_{DP}) < 100\ mm\ h^{-1}$. However, an in-depth analysis reveals that the magnitudes of rainfall accumulations are different at higher rain rates. From 16:00 to 17:00 LST (the right column in Fig. 8), the rainfall hot spots are in the vicinity of the Zhengzhou radar site (black triangle Fig. 8). The LP method is characterized by the largest area of $R(K_{DP}) > 130\ mm\ h^{-1}$ (Fig. 8$a_3$), while the smallest area was found for the Maesaka algorithm (Fig. 8$b_3$). However, due to the scaricity of gauges in the area of rainfall hot spots, this difference is noticeable only for the gauge with the 201.9 $mm\ h^{-1}$ report (black cross Fig. 8).

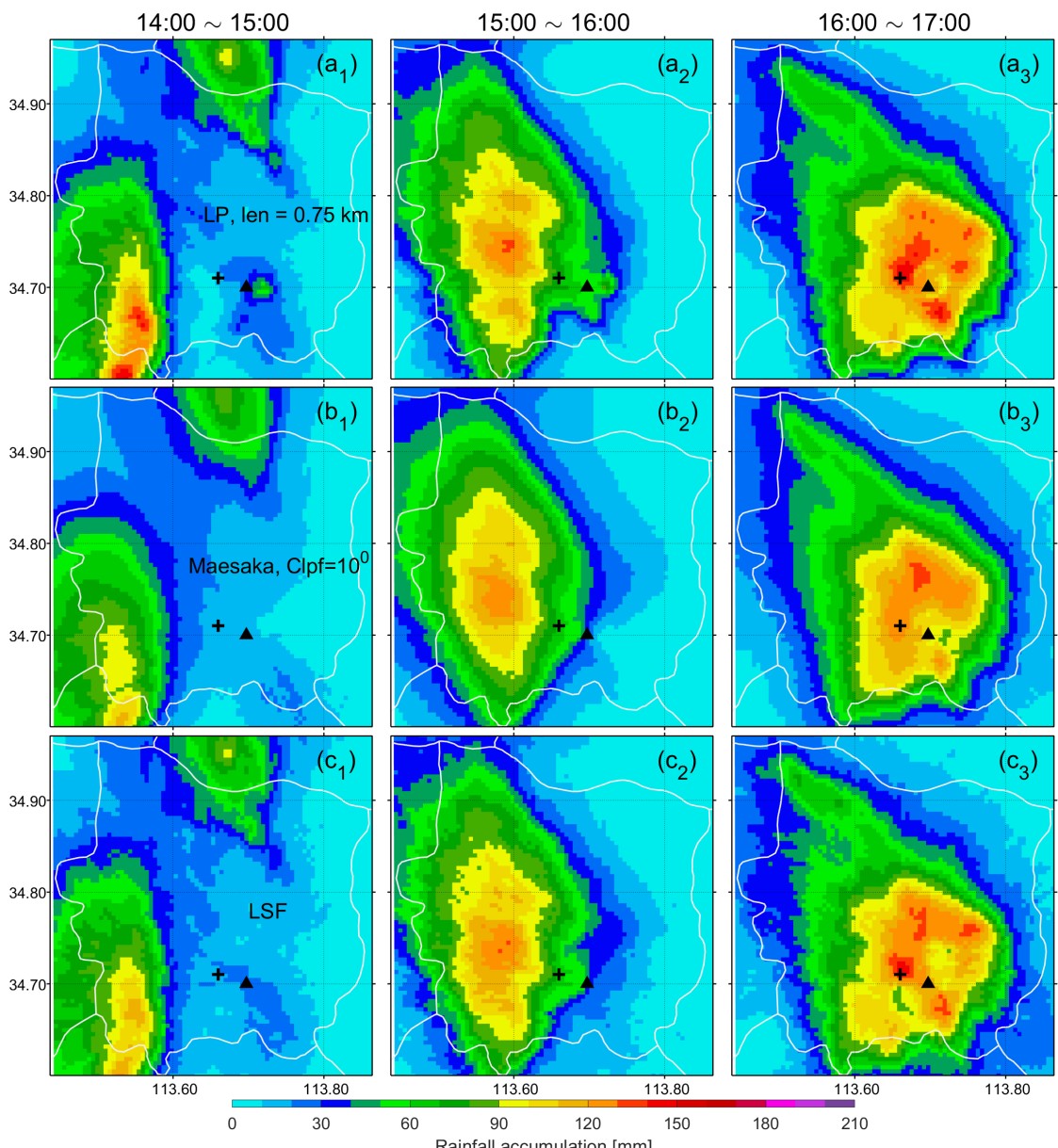

**Figure 8.** Hourly areal rainfall map from 14:00 to 17:00 LST. $K_{DP}$ was estimated from the (a) LP method with LP = 0.75 km, (b) Maesaka method with Clpf = $10^0$, and (c) LSF method. The black triangle and cross denote Zhengzhou radar and the site hosting the gauge with the 201.9 $mm\ h^{-1}$ report, respectively. $R = 51 K_{DP}^{0.86}$ was used.

The areal hourly rainfall accumulation enables the analysis of the evolution of this event. As shown in Fig. 8a, the precipitation system moved into Zhengzhou city from the southwest pouring rainfall up to 130 $mm\ h^{-1}$ from 14:00 to 15:00 LST (Fig. 8a). Then it slowly propagated northeastwards in the next one hour with increased precipitation intensity. The hourly rainfall

beyond 100 $mm\ h^{-1}$ covered a north-south oriented, ellipse-shaped area of about 115.5 $km^2$. From 16:00 to 17:00 LST, the precipitation system moved eastwards and poured the most intense hourly rainfall over the center of Zhengzhou city (Fig. 8c). The rainfall rate beyond 100 $mm\ h^{-1}$ covered an area of about 198.25 $km^2$, which is 171.7% of that in the previous one hour. The increased rainfall extremity and the more localized extreme rainfall likely resulted from merging of convective cells and formation of an arc-shaped convergence zone which favored the development of convective updrafts in a three-quarter circle around the storm (Yin et al., 2022). Interestingly, the gauge with the 201.9 $mm\ h^{-1}$ report was almost exactly located in the high-value center of the hourly rainfall map at 17:00 LST.

The accumulated rainfall from 14:00 to 17:00 LST is presented in Fig. 9. As expected, the results of the LP method and the LSF method are similar, while the area of rainfall accumulation exceeding 200 $mm$ generated by the Maesaka method is significantly different from those using the other two methods. Interestingly, we have found that the center of 3-h rainfall accumulation was off from the hot spot with the record-breaking hourly rainfall accumulation (16:00 ~ 17:00 LST, Fig. 8a$_3$). Specifically, the center of 3-h rainfall accumulation was located southwest of Zhengzhou city, fortunately an urban-rural fringe area where the surface is less impervious and relatively fewer residents were living.

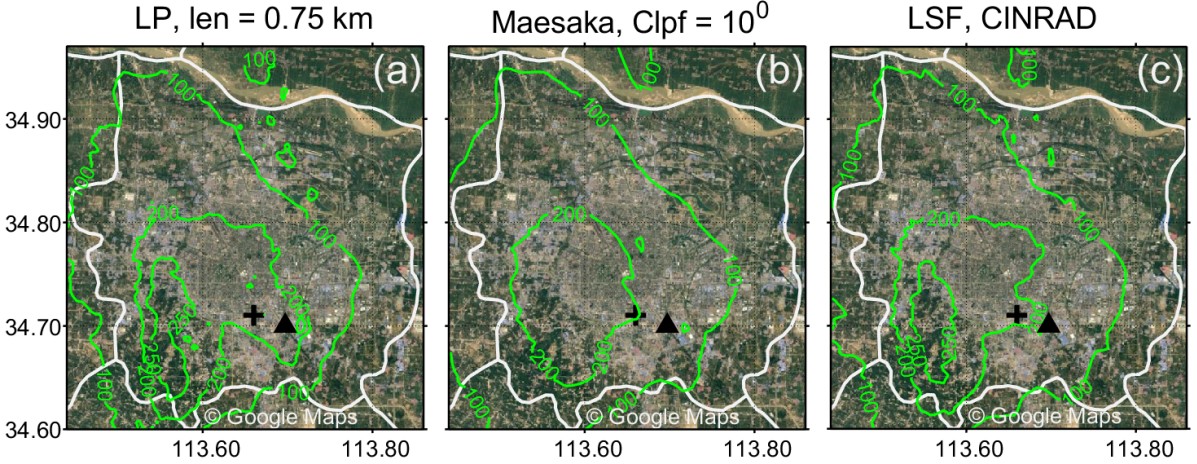

**Figure 9.** Satellite images from Google Maps overlapped by isolines indicating the rainfall accumulation [mm] during 14:00 ~ 17:00 LST. The rain rate was inferred from $R = 51 K_{\mathrm{DP}}^{0.86}$ in which $K_{\mathrm{DP}}$ was estimated from the (a) LP method with len = 0.75 km, (b) Maesaka algorithm with Clpf = $10^0$, and (c) LSF method. The black triangle and cross denote Zhengzhou radar and the site hosting the gauge with the 201.9 $mm\ h^{-1}$ report, respectively.

## 6 Conclusions

In this study, we have examined the $K_{DP}$-based QPE for the record-breaking extreme rainfall event occurred over Zhengzhou, 14:00 $\sim$ 17:00 20 July 2021 LST. The rain drop size distribution observations obtained by an OTT disdrometer was used to develop $R(K_{DP})$ parameterizations. The $K_{DP}$ estimates generated by operationally used LSF method were compared with two parameter-controlled methods. The $K_{DP}$ estimates were gridded with a spatial resolution of 500 m and the results of $R(K_{DP})$ were compared with gauge observations. The results can be summarized as follows.

– Range degradation effect significantly affected the performance radar-based QPE in this event. The precipitation center as identified by the Luoyang radar, which is about 120 km from the Zhengzhou city center, significantly deviates from Zhengzhou radar estimates.

– The assumed $\sigma$ in T-matrix simulation has tangible impact on the development of $R(K_{DP})$ parameterizations. Higher $\sigma$ results in smaller $K_{DP}$ in simulations for a given rain drop size distribution. The previous Bringi et al. (2008) experimental study on $\sigma$ was made in low-wind conditions, while the applicability of $\sigma$ assumption in moderate to strong winds should be addressed in future studies.

– Gauges deployed over the Zhengzhou city were used to evaluate the accuracy of $R(K_{DP})$. The results show that all methods agree with each other rather well for $R(K_{DP}) < 100\ mm\ h^{-1}$. The LP method is capable of producing the highest rainfall accumulation. In a statistical sense, changing the window length from 0.75 to 11.25 km in the LP method or Clpf from $10^0 \sim 10^4$ in the Maesaka algorithm does not significantly affect the QPE performance, while the oversmoothing was found for the Maesaka algorithm with Clpf=$10^6$.

– $K_{DP}$ estimates of three algorithms over the gauge with the 201.9 $mm\ h^{-1}$ report were compared, and the results are generally similar to (Reimel and Kumjian, 2021). One notable difference is that the estimated $K_{DP}$ almost reached "saturation" at the window length of 3.75 km, and the increase of $K_{DP}$ with the decrease of window length is not as significant as that in (Reimel and Kumjian, 2021). The results of LP method with a window length of 0.75 km are close to those of the LSF method, but significantly larger than the highest values obtained from the Maesaka algorithm.

– $R(K_{DP})$ with the $K_{DP}$ estimated from the three methods cannot reproduce the gauge-observed 201.9 $mm\ h^{-1}$. Our comparisons suggest that this underestimation is unlikely attributed to the $K_{DP}$ estimation process. Rather, the adequacy of assumed $\sigma$ and unquantified vertical air motions may explain this underestimation.

– The gauge with the 201.9 $mm\ h^{-1}$ report was located at the vicinity of local rainfall hot spots during 16:00 $\sim$ 17:00 LST, but the center of the 3-h areal rainfall accumulation was found to be located at the southwest of Zhengzhou city, deviating from the site with the 201.9 $mm\ h^{-1}$ record.

From the perspective of operational applications, the effect of smoothing on $K_{DP}$ estimation is interesting. Our results show that the use of smoothing factor has minimal impact on $K_{DP}$ for hourly rainfall accumulation below 100 mm, while its impact

becomes more significant as the rain rate increases. This suggests the importance of employing an adaptive window length as used in the LSF method. However, current LP or Maesaka algorithm uses a fixed window length or a single smoothing factor. It is recommended to develop a new LP algorithm with an adaptive window length in the future. In addition, we speculate that the underestimation of 201.9 $mm\ h^{-1}$ rainfall accumulation can be attributed to the inadequate assumptions about raindrop microphysics and unquantified vertical air motions. Although we cannot quantify their contributions in the Zhengzhou event, more focused observational experiments are suggested to ascertain their impact on radar-based QPE.

Extreme rainfall events are relatively rare, but they are very destructive. We call for integrated efforts to tackle the issue of radar data quality control, and to promote the capability of operational weather radars in extreme rainfall monitoring. This will improve hydrological modelling, extreme rainfall nowcasting and disaster mitigation for cities, and will also be valuable to the studies of mechanisms governing the extreme rainfall production.

*Data availability.* The data used in this study can be accessed by contacting the first author. The merged $K_{DP}$ figures are availble at https://github.com/HaoranLiHelsinki/Figs_Zhengzhou . The hourly QPE products generated in this study are availble at https://github.com/HaoranLiHelsinki/QPE_zhengzhou.

*Author contributions.* HL and DM conceptualized the study. HL performed the experiment and wrote the paper. All the authors took part in the interpretation of the results and edits of the paper.

*Competing interests.* The authors declare that they have no conflict of interest.

*Acknowledgements.* We thank Dr. Scott Giangrande, Dr. Zhe Zhang and Dr. Tanel Voormansik for helpful discussions on the linear programming method. This research has been supported by the National Natural Science Foundation of China (grants no. 42030610, U2142210) and the Basic Research & Operation Fund of CAMS (451490).

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
