# Peer review of "Assessing $K_{\rm DP}$ -based QPE for the record-breaking rainfall over Zhengzhou city on 20 July 2021"

_Hydrology and Earth System Sciences, 2022_

## Author Comment (AC1)

**Reviewer #1**

The study addresses polarimetric radar estimation of extreme rainfall using specific differential phase KDP. The performance of different versions of the KDP-based estimators is compared and tested using rainfall gauge measurements. The authors consider various KDP processing techniques and possible impacts of the DSD variability and raindrop orientations uncertainty on the quality of the polarimetric rainfall estimation.

It is demonstrated that all tested methods optimized by a locally measured DSD in a particular heavy rain event tend to yield reasonably good estimates of rainfall with hourly totals below 100 mm but significantly underestimate heavier rain with rain rates approaching 200 mm/h. The authors speculate that such underestimation can be attributed to a more random orientations of raindrops than is commonly assumed in the derivation of the R(KDP) relation or to a big difference between the radar and gauge sampling volumes.

This reviewer believes that the authors may underestimate the impact of downdrafts which are often associated with a torrential rain. They downplay such an impact arguing that it is not detected by the disdrometer. The fact is that the vertical air motion is always negligible near the surface which does not exclude high downdraft speeds just a few hundred meters above the surface where the radar samples rainfall. Under such scenario, radar underestimation of rain is inevitable because of the conservation of a precipitation flux. The authors have a unique chance to check this hypothesis. Of course, direct measurements of the vertical air velocity are usually not available with the operational radars but one can examine indirect radar attributes of downburst such as strong wind divergence near the surface which can be assessed from the Doppler measurements or rapidly descending KDP or Z cores that are proven to be linked to the downbursts / microbursts.

We would like to thank the reviewer for very good suggestions which helped us to refine this work. In the revised manuscript, we have added a figure showing the time serials of Doppler velocity and the radial divergence over the Zhengzhou national climatological station. The results suggest sustained divergence from 16:00 to 17:00 over the site reporting the 201.9 mm report. Therefore, we agree that the existence of downdrafts should not be ruled out. But we did not see significant descending $K_{DP}$ or Z in our observations, which may be explained by the limited sampling heights (0.083 km, 0.132 km, 0.182 and 0.237 km at 1.5°, 2.4°, 3.3° and 4.3°). Please see the details in the revised manuscript.

I have several other recommendations and concerns.

1.  What is the purpose of comparing the data from the Luoyang and Zhengzhou radars? Only the latter radar data are used for a qualitative analysis. Once the maps of rain totals retrieved from the two radar are displayed in Fig. 2, the reasons for discrepancies

have to be discussed. Most likely, an apparent shift in the areas of heaviest rainfall is related to the differences in the altitudes of the radar sampling volume and strong vertical gradients of rain rates which are typical for warm rain process.

The reviewer is correct. In the revised manuscript, we have made more detailed discussions on this point.

*Yin et al. (2022) have made simulations on this event, and they found that the storms were tilted eastward. The sampling volume of the Luoyang radar over Zhengzhou city is about 2 km, while the Zhengzhou radar observes near-surface precipitation. Therefore, the precipitation observed by the Luoyang radar is more eastward than the Zhengzhou radar. In addition, warm rain processes may also significantly augment rain rates within the height of 2 km (Yu et al., 2022). Given the effects discussed above, Zhengzhou radar will be used for QPE in this study.*

2. Self-consistency factor and Clpf should be defined.

   Both are tunable coefficients in Py-ART codes to impose the weights on final solutions. We have added the following information into the algorithm introduction section.

   *In Py-ART, the self-consistency factor is used to define the weight of the $Z_e$-$K_{DP}$ relationship on the final solution, and the default value is $6*10^4$.*

   *The effect that the low pass filter has on the final solution depends on a user-defined parameter Clpf. By changing the value of Clpf the user can control the amount of smoothing applied by the algorithm.*

3. Averaging window lengths (len) have to be expressed in km.

   Agree, we have amended the manuscript as suggested.

4. Keep in mind that the LP procedure always tend to overestimate KDP because it ignores negative radial slopes of ΦDP. Negative KDP is always coupled with overestimated positive KDP in the close proximity and both shouldn't be quantitatively used. This is a nature of the nonuniform beam filling impact on the KDP.

   The reviewer is correct. But this effect does not seem to be significant in this study. We have added the discussion about this point in the revised manuscript:

   *It should be noted that the non-uniform radar beam filling was not considered in idealized known-truth tests (Reimel and Kumjian, 2021), but it can lead to local perturbation of $K_{DP}$ (Ryzhkov and Zrnic, 1998). Because the LP and Maesaka et al. (2012) methods assume the monotonic increase of ΦDP, and therefore they are expected to yield higher $K_{DP}$ than the LSF method if the negative radial slope of ΦDP occurs in the close proximity. This effect does not seem to be significant in this study for the following reasons. Firstly, the Zhengzhou radar is close to the gauge site (3.15 km), and therefore the radar sampling volume is much smaller than that at larger distances. Then, the gauge site was not located in the edges of rain cells (see $K_{DP}$ composites at https://github.com/HaoranLiHelsinki/Figs_Zhengzhou). Finally, we have manually checked ΦDP observations, and did not see significant negative radial slope*

*of ΦDP. In addition, the smallest Clpf (least smoothing) yields smaller $K_{DP}$ than the LSF method from 16:00 to 17:00 LST (Fig. 3a), suggesting the selection of $K_{DP}$ estimation method is more important than the effect of non-uniform radar beam filling in this study.*

5.  I don't see any difference between the three panels in Fig. 8, It is not clear what is displayed because the shading is not specified.

    We feel sorry that the explanation about the isolines was missing. We have amended the first sentence of the caption as,

    *Satellite images from Google Maps overlapped by isolines indicating the rainfall accumulation [mm] during 14:00 ~ 17:00 LST.*

English usage has to be substantially improved. Just a few examples follow

We thank the reviewer for correcting the grammar mistakes. We have made careful proofreading for the revised manuscript, please see the revised manuscript.

L 32. Should be "coefficient" instead of "ratio"

Corrected.

L 60. Remove "the" before "progress"

Corrected.

L 72. KDP is a radial derivative (not derivation)

This sentence has been amended as:

*$K_{DP}$ is one-half the range derivative of differential phase shift.*

L 120. This "may be attributed"

Corrected.

L 133. "A linear interpolation"

Corrected.

L 153. "At 274°"

Corrected.

L 174. KDP is less dependent on DSDs than what?

This sentence has been amended as:

*$K_{DP}$ is less dependent on DSDs than other radar products.*

L 196. "Earlier" instead of "early"

Corrected.

L 217. "Analyze" instead of "analysis"

Corrected.

L 222. Replace "decent"

"Decent" has been replaced by "detectable".

L 278. "In-depth" instead of "depth-in"

Corrected.

L 298. Remove "at the"

Corrected.

L 304. Gridded

Corrected.

L 309. Replace "decent" with "significant" or "noticeable"

Corrected.

---

## Author Comment (AC2)

**Reviewer #2**

**Summary**: This concise paper is an important application and demonstration of KDP estimation uncertainty for a devastating flooding case in China. The authors perform a robust statistical analysis of the different parameter choices for popular KDP estimation algorithms, compared to the operational algorithm for the CINRAD radar networks. This type of work has been done for synthetic observations, but not for real cases with such dense ground-based (gauge) observations for evaluation.

The paper is free of any major fatal flaws. However, there are a large number of mostly minor comments that need to be addressed. One structural comment -- there is not really a good concluding paragraph (see the first comment below). Additional proofreading is necessary. For these reasons, I suggest MAJOR revisions, although this is somewhere between major and minor.

We sincerely appreciate the reviewer for constructive comments on our paper. We have amended the manuscript as suggested. Please see below our response to your comments.

**Comments**:

1. The conclusion section is a nice summary of the study, but it sort of ends abruptly without wrapping up. The authors need to include a concluding paragraph (it can be brief), that brings the focus back out to the broader perspective. This is sometimes referred to as the "funnel technique" or structure. What do the results of this study contribute to the community's knowledge or application of rainfall estimation? What do you recommend for future work, and how will your efforts contribute to the main goal of mitigating losses from devastating flooding events? Some answers to those questions are needed.

We thank the reviewer for the good suggestion. We have added a paragraph discussing the implications of this study as follows,

*From the perspective of operational applications, the effect of smoothing on $K_{DP}$ estimation is interesting. Our results show that the use of smoothing factor has minimal impact on $K_{DP}$ for hourly rainfall accumulation below 100 mm, while its impact becomes more significant as the rain rate increases. This suggests the importance of employing an adaptive window length as used in the LSF method. However, current LP or Maesaka algorithm uses a fixed window length or a single smoothing factor. It is recommended to develop a new LP algorithm with an adaptive window length in the future. In addition, we speculate that the underestimation of 201.9 mm $h^{-1}$ rainfall accumulation can be attributed to the inadequate assumptions about raindrop microphysics and unquantified vertical air motions. Although we cannot quantify their contributions in the Zhengzhou event, more delicate observational experiments are suggested to ascertain their impact on radar-based QPE.*

2. L27: "seem falling short" should be "seem to be falling short"?

Corrected.

3. L31: What is meant by "parameterized reflectivity factor"? That is not standard usage, unless it means something different from the traditional equivalent radar reflectivity factor?

'Parameterized reflectivity factor' has been replaced by 'equivalent radar reflectivity factor'.

4. L32: remove "the" before "attenuation effects"

Corrected.

5. L43: "infrastructures" should be "infrastructure"

Corrected.

6. L67: Maybe a reference for KDP being immune to beam blockage would be helpful to readers less familiar with the dual-pol products.

(Lang et al., 2007) has been added in the revised manuscript.

7. L69: "data was" should be "data were". Same in L134.

Corrected.

8. L72: "derivation" should be "derivative". Also, maybe adding "derivative with respect to range" or something like that to clarify?

Agree. This sentence has been amended as:

*$K_{DP}$ is one-half the range derivative of differential phase shift.*

9. L78: The Maesaka reference should not be in parentheses according to the style guide…same in L147, L169, and elsewhere; also, no capitalization needed for "Linear"

Corrected. Specifically, the algorithm developed by Maesaka (2012) is referred as Maesaka algorithm in the revised manuscript.

10. L98: are there units for these numbers? Are they range gates?

Yes, they are range gates. In the revised manuscript, they have been expressed in km.

11. L105: Is there an objective routine for the removal of spikes, or is this done manually? (Either is fine, just indicate how it is done for reproducibility)

Both were done for ensuring the data quality of $\Phi_{DP}$. In the revised manuscript, the following has been added.

*To minimize the impact of those spikes on KDP estimation, the following procedures were made:*

*– Firstly, a linear fit was made to the raw $\Phi_{DP}(r)$ data for an interval of 5 km. The fitted values were labeled as $\Phi'_{DP}(r)$.*

*– Then, the point with $|\Phi_{DP}(r)-\Phi'_{DP}(r)|$ exceeding 10° was identified as clutter.*

*– Finally, a cubic spline interpolation was made to the identified clutter points.*

*These steps can effectively remove majority of clutter signals, however, local perturbation of $\Phi_{DP}$ can be on the order of 10° given the area of interest is so close to the radar. Therefore, we have also manually checked the $\Phi_{DP}$ fields and removed significant clutter signals.*

12. L117: Is there a threshold used to define "significantly deviating"?

In the revised manuscript, this part has been amended as follows.

*For the HYDRO gauges with hourly measurements, the inverse distance weighting (IDW) approach (Chen and Liu, 2012) was implemented to yield an estimate of hourly rainfall accumulation at a given HYDRO gauge site. Then, the observed value below 50% of the expected one was removed. This method was mainly used for identifying gauges which were not working due to power outages.*

13. L122: "This may attribute" should be "This may be attributed to"

Corrected.

14. L154: "elevation angle dependence of KDP" – this confused me at first. I believe the authors are referring to the viewing angle of raindrops effect, and not any sort of vertical profile of KDP effects? Maybe some sort of clarification would help.

In the revised manuscript, this part has been amended as follows.

*Here, radar observations at the elevation angles of 1.5°, 2.4°, 3.3° and 4.3° were used for the following considerations. (1) The dependence of observed KDP on the viewing angle is expected to be negligible at small radar elevation angles, i.e., smaller than 4.3° (Bringi and Chandrasekar, 2001); (2) Given the strong ground clutter contamination, we discarded the data recorded at the lowest elevation angle and $K_{DP}$ estimates at elevation angles of 1.5°, 2.4°, 3.3° and 4.3° corresponding to heights about 0.083 km, 0.132 km, 0.182 km and 0.237 km, respectively, over the station were used.*

15. L223: the word "cry" is not correct, and I don't know what is being stated.

This sentence has been amended as:

*With a larger σ (Fig. 5b), $R(K_{DP})$ is still well below OTT/gauge observations.*

16. L234-238: There shouldn't be any downdrafts experienced at the surface owing to mass continuity (i.e., downdrafts become diverging outflow at the surface). So, this argument does not make physical sense. There almost certainly will be downdrafts above this level, perhaps in the region where the radar is sampling. Are there any low-level

divergence observations from the radar or surface stations that could be used to estimate the downdraft speed at radar beam height? (This is admittedly crude, but would help indicate if the scale of the downdraft is significant or not.)

The reviewer raised a very good point. In the revised manuscript, we have added a figure showing the time serials of Doppler velocity and the radial divergence over the Zhengzhou national climatological station. The results indicate sustained divergence from 16:00 to 17:00 over the site reporting the 201.9 mm report. Therefore, we agree that the existence of downdrafts should not be ruled out.

17. L242-243: I'm not sure 6-10 m/s winds would be consistent with a "tornado." Is this a misunderstanding of the cited reference? In other words, did the storm produce a tornado, but the sampled winds were obviously much weaker? Please clarify.

We have clarified this point in the revised manuscript. Firstly, it is 'a tornadic squall-line storm'; Then, the wind speed was '4-min running averaged'.

18. L246-248: This seems really unlikely, in my opinion. Are there any references to support this conjecture about extremely narrow heavy rain shafts? This is important because it is alluded to in one of your conclusions (L325).

We have rewritten this part in the revised manuscript as follows, and the corresponding sentences in conclusions have been deleted.

*Although this effect is difficult to quantify, we argue that it plays a minor role for the rainfall underestimation. By manually checking the movement of storms (merged $K_{DP}$ observations at https://github.com/HaoranLiHelsinki/Figs_Zhengzhou, we found that the storm propagation speed is on the order of several kilometers per hour, contrasting with the much smaller radar sampling volume. Given the rapid changing nature of the storms, the sampling effect does not seem to be a major factor responsible for the rainfall underestimation.*

19. L258: "imposes obviously oversmoothing filter" should be revised, maybe something like "imposes an overaggressive filter that obviously leads to oversmoothing"?

Corrected.

---

## Referee Report (RR1)

Comments to the Author

Review of the paper "Assessing $K_{DP}$-based QPE for the record-breaking rainfall over Zhengzhou city on 20 July 2021".

In this paper, for the record-breaking rainfall in Zhengzhou City on 20 July 2021, the performance of $K_{DP}$-based quantitative precipitation estimation is evaluated based with ground observation equipments (rain gauge, disdrometer and S-band dual-polarization radar). Several "best $K_{DP}$ estimate" and "Parameterizations of $R(K_{DP})$" methods are discussed in depth. On this basis, the quantitative precipitation estimation results based on $K_{DP}$ are analyzed in the case of single-point and multi-point statistical rain rate, and areal rainfall maps, respectively. Finally, the research work is concluded.

The logic of this paper is clear and rigorous, and the discussion and analysis are in-depth and convincing after the previous review process. It is meaningful for the application of polarimetric radar in the monitoring of extreme precipitation and the corresponding disaster warning. However, there are also some deficiencies in the writing of the paper, and this paper can be published after these are resolved. Therefore, my suggestion is minor revision.

Below are some specific comments.
Major comments:
1. Extreme rainfall events are rare but destructive, so it is important to monitor them indeed. Only one extreme rainfall event was studied in this study, and whether the robustness of some conclusions obtained needs to be further tested, which may be important to assess the strength of the significance of this study. Of course, I understand that extreme rainfall events are difficult to capture, and the title of this paper is aimed at this extreme rainfall event over Zhengzhou city on 20 July 2021. However, considering the robustness of $K_{DP}$-based QPE algorithm is meaningful for practical applications. Therefore, I suggest that the author can mention relevant content in the later part of the paper. It's not mandatory.
2. Line 251-252: I'm wondering if it can deduce this conclusion based on the results. The fact that one method is better than the other two does not seem to mean that the uncertainty of that method has been minimized. Perhaps none of the three methods is ideal for achieving functionality. After all, for such an empirically fitted model, it is more likely to be more about convenience in the operation system.

Minor comments:
1. Figure 1(d): This figure is not mentioned in the paper, although the relevant wind speed information is explained later in the paper.
2. Line 62: There are some inconsistent expressions, such as "Fig. 1(a)", "Figure 1(c)", "Fig. 2b", please unify them in the full text.
3. Line 62-63: Latitude and longitude keep the same number of digits after the decimal point.

4. Line 78: The references here should not be bracketed, please revise it in full text scope.

5. Line 128: OTT PARSIVEL's version needs to be noted. As far as I know, there is a difference in accuracy between the first and second generation.

6. Line 154: The author only used Zhengzhou radar for QPE. The Luoyang radar seems only to have been used to show the difference in the location of heavy rainfall centers at different heights as shown in Figure 2. However, the purpose of the article is "Assessing KDP-based QPE for the record-breaking rainfall over Zhengzhou city on 20 July 2021". Although it is interesting in revealing the phenomenon, I am considering the necessity of radar related content in Luoyang in this paper. The author may consider my opinion, but it is not mandatory.

7. Figure 2: "$R=51K_{DP}^{0.86}$" needs to be stated here, even if mentioned in Section 3.2.

8. Line 180: I'm not sure if "noise" is appropriate, perhaps uncertainty?

9. Line 204: Replace "16~17" with "16:00~17:00", please revise it in full text scope.

10. Line 206: Replace "well.Therefore" with "well. Therefore".

11. Line 258-259: Explaining why or adding references is needed.

12. Section 4.2: Does the rain gauge here include meteorology and hydrology? How many in total?

13. Line 296: std might be better in italics.

14. Figure 7: Please refine this drawing. Different subgraphs appear overwritten, and (e) the legend of the graph is out of the graph box.

15. Line 361: Replace "saturation" with "saturation".

---

## Author Response (AR2)

**Reviewer #1**

The authors need to change the way they use parenthesis in the citations of other papers in the manuscript. I assume that it is more appropriate to use "the method of Maesaka et al. (2012)" rather than "the method of (Maesaka et al. 2012)" (in line 80 and several other places).

Corrected.

L 112: Should be "procedures were utilized" instead of "procedures were made"

Corrected.

L 123: Remove a hyphen in "gauge-observations"

Corrected.

L. 158: "Simulations of this event"

Corrected.

L. 315: Use "too aggressive" instead of "overaggressive"

Corrected.

L. 368: It is better to use "tangible" or "noticeable" instead of "detectable"

Corrected.

L. 394: Use "more focused" instead of "more delicate"

Corrected.

**Reviewer #3**

In this paper, for the record-breaking rainfall in Zhengzhou City on 20 July 2021, the performance of KDP-based quantitative precipitation estimation is evaluated based with ground observation equipments (rain gauge, disdrometer and S-band dual-polarization radar). Several "best KDP estimate" and "Parameterizations of R(KDP)" methods are discussed in depth. On this basis, the quantitative precipitation estimation results based on KDP are analyzed in the case of single-point and multi-point statistical rain rate, and areal rainfall maps, respectively. Finally, the research work is concluded. The logic of this paper is clear and rigorous, and the discussion and analysis are in-depth and convincing after the previous review process. It is meaningful for the application of polarimetric radar in the monitoring of extreme precipitation and the corresponding disaster warning. However, there are also some deficiencies in the writing of the paper, and this paper can be published after these are resolved. Therefore, my suggestion is minor revision. Below are some specific comments.

Major comments:

1. Extreme rainfall events are rare but destructive, so it is important to monitor them indeed. Only one extreme rainfall event was studied in this study, and whether the robustness of some conclusions obtained needs to be further tested, which may be important to assess the strength of the significance of this study. Of course, I understand that extreme rainfall events are difficult to capture, and the title of this paper is aimed at this extreme rainfall event over Zhengzhou city on 20 July 2021. However, considering the robustness of KDP-based QPE algorithm is meaningful for practical applications. Therefore, I suggest that the author can mention relevant content in the later part of the paper. It's not mandatory.

We have added a paragraph in conclusions.

*Extreme rainfall events are relatively rare, but they are very destructive. We call for integrated efforts to tackle the issue of radar data quality control, and to promote the capability of operational weather radars in extreme rainfall monitoring. This will improve hydrological modelling, extreme rainfall nowcasting and disaster mitigation for cities, and will also be valuable to the studies of mechanisms governing the extreme rainfall production.*

2. Line 251-252: I'm wondering if it can deduce this conclusion based on the results. The fact that one method is better than the other two does not seem to mean that the uncertainty of that method has been minimized. Perhaps none of the three methods is ideal for achieving functionality. After all, for such an empirically fitted model, it is more likely to be more about convenience in the operation system.

As shown by Reimel and Kumjian (2021), by varying the control parameters in LP and Maesaka algorithms the estimated KDP changes from overestimation to underestimation.

In this study, we show that even though with the highest estimated KDP, we could not obtain the in situ observations. We have amended this sentence as

*Therefore, we should have good confidence that the best $K_{DP}$ should be close to or lower than the estimates.*

Minor comments:

1. Figure 1(d): This figure is not mentioned in the paper, although the relevant wind speed information is explained later in the paper.

We have added (Fig. 1d) in the revised manuscript.

*However, the observed horizontal wind speed is 3 ~ 5 m s$^{-1}$ from 16:00 to 17:00 LST (Fig. 1d).*

2. Line 62: There are some inconsistent expressions, such as "Fig. 1(a)", "Figure 1(c)", "Fig. 2b", please unify them in the full text.

Corrected.

3. Line 62-63: Latitude and longitude keep the same number of digits after the decimal point.

Corrected.

4. Line 78: The references here should not be bracketed, please revise it in full text scope.

Corrected.

5. Line 128: OTT PARSIVEL's version needs to be noted. As far as I know, there is a difference in accuracy between the first and second generation.

We have added the following in 2.2:

*The one deployed close to the gauge is the second generation of OTT.*

6. Line 154: The author only used Zhengzhou radar for QPE. The Luoyang radar seems only to have been used to show the difference in the location of heavy rainfall centers at different heights as shown in Figure 2. However, the purpose of the article is "Assessing KDP-based QPE for the record-breaking rainfall over Zhengzhou city on 20 July 2021". Although it is interesting in revealing the phenomenon, I am considering the necessity of radar related content in Luoyang in this paper. The author may consider my opinion, but it is not mandatory.

We have decided to keep this figure since which radar should be used for QPE is also a relevant topic to address.

7. Figure 2: "R=51KDP0.86 " needs to be stated here, even if mentioned in Section 3.2.

We have added this to the caption of Figure 2.

8. Line 180: I'm not sure if "noise" is appropriate, perhaps uncertainty?

Corrected.

9. Line 204: Replace "16~17" with "16:00~17:00", please revise it in full text scope.

Corrected.

10. Line 206: Replace "well.Therefore" with "well. Therefore".

Corrected.

11. Line 258-259: Explaining why or adding references is needed.

$K_{DP}$ is indicative of DSD changes, but we do not see significant changes of $K_{DP}$ as explained in the text.

In the revised manuscript, we have added a reference supporting that the rainwater content does not change significantly within 200-300m.

12. Section 4.2: Does the rain gauge here include meteorology and hydrology? How many in total?

We have added the explanation:

*The dense meteorological and hydrological rain gauge network in Zhengzhou city allows a statistical evaluation of the KDP-based QPE.*

We have added the following sentence in section 3.2:

*After the data quality control, 114 gauges were used in this study.*

13. Line 296: std might be better in italics.

Corrected.

14. Figure 7: Please refine this drawing. Different subgraphs appear overwritten, and (e) the legend of the graph is out of the graph box.

Corrected.

15. Line 361: Replace "saturation" with "saturation"

Corrected.